# Water Security Assessment of the Grand River Watershed in Southwestern Ontario, Canada

**Baljeet Kaur [1], Narayan Kumar Shrestha [1], Prasad Daggupati [1,*], Ramesh Pal Rudra [1], Pradeep Kumar Goel [2] , Rituraj Shukla [1] and Nabil Allataifeh [1]**

[1] School of Engineering, University of Guelph, Guelph, ON N1G 2W1, Canada; baljeet@uoguelph.ca (B.K.); shresthn@uoguelph.ca (N.K.S.); rrudra@uoguelph.ca (R.P.R.); rshukla@uoguelph.ca (R.S.); nallatai@uoguelph.ca (N.A.)

[2] Ontario Ministry of the Environment, Conservation and Parks, Etobicoke, ON M9P 3V6, Canada; Pradeep.Goel@ontario.ca

\* Correspondence: pdaggupa@uoguelph.ca

**Abstract:** Water security is the capability of a community to have adequate access to good quality and a sufficient quantity of water as well as safeguard resources for the future generations. Understanding the spatial and temporal variabilities of water security can play a pivotal role in sustainable management of fresh water resources. In this study, a long-term water security analysis of the Grand River watershed (GRW), Ontario, Canada, was carried out using the soil and water assessment tool (SWAT). Analyses on blue and green water availability and water security were carried out by dividing the GRW into eight drainage zones. As such, both anthropogenic as well as environmental demand were considered. In particular, while calculating blue water scarcity, three different methods were used in determining the environmental flow requirement, namely, the presumptive standards method, the modified low stream-flow method, and the variable monthly flow method. Model results showed that the SWAT model could simulate streamflow dynamics of the GRW with 'good' to 'very good' accuracy with an average Nash–Sutcliffe Efficiency of 0.75, $R^2$ value of 0.78, and percentage of bias (PBIAS) of 8.23%. Sen's slope calculated using data from over 60 years confirmed that the blue water flow, green water flow, and storage had increasing trends. The presumptive standards method and the modified low stream-flow method, respectively, were found to be the most and least restrictive method in calculating environmental flow requirements. While both green (0.4–1.1) and blue (0.25–2.0) water scarcity values showed marked temporal and spatial variabilities, blue water scarcity was found to be the highest in urban areas on account of higher water usage and less blue water availability. Similarly, green water scarcity was found to be highest in zones with higher temperatures and intensive agricultural practices. We believe that knowledge of the green and blue water security situation would be helpful in sustainable water resources management of the GRW and help to identify hotspots that need immediate attention.

**Keywords:** water security; blue water; green water; Grand River watershed; soil and water assessment tool (SWAT)

## 1. Introduction

Water security is the capability of a community to have adequate access to good quality and a sufficient quantity of water as well as safeguard resources for the future generations [1]. The seemingly implacable rise in water demand, limited water supply, demographic changes, high standards of living, and uneven distribution of water resources are some of the factors that have led to a rise in water security issues all around the world [2]. Many rivers around the world are running dry for

considerable parts of the year before merging into the sea. Not only the surface water resources, but also the ground water is being overly exploited with pumping rates faster than replenishment, thus increasing threats to the availability of freshwater resources for irrigation, human consumption, energy production, and environmental sustainability [3,4]. It has been reported that 1.8 billion people are likely to be at high risk for water scarcity by 2025 [5,6]. Thus, a thorough understanding of the spatial and temporal variabilities of water security is required to sustain the available fresh water resources [7].

One method for dealing with water scarcity is classifying the freshwater resources into blue and green water [4] for more specific and targeted managements of each. Blue water (BW) is generally considered as the water flowing on the surface and through sub-surface courses, stored in rivers, lakes, and deep aquifers, while green water (GW) is characterized as the water stored in unsaturated soil layers and water moving to the atmosphere through transpiration and evaporation [8,9]. Green water plays a critical role in agricultural crop production and accounts for nearly 80% of the total water used for global crop production [10]. In tropical regions where rain-fed agriculture plays a major role, primary water usage is that of green water. Knowing the quantity and specific consumption of available blue and green water can help in sustainable management of these water resources.

Similarly, a water footprint defines the link between human water use and a growing population [4]. BW footprint is the consumptive use of surface and ground-water resources for producing goods and services, and GW footprint is the green water consumption (evapotranspiration) that directly or indirectly (agriculture) benefits humans. The disparity between water use and water availability can be characterized by various methods (i.e., shortage, stress, and scarcity), and one of the globally used methods is to estimate water scarcity [4,8,9] as the ratio of water footprint to available water. Assessment of water scarcity is one of the prerequisites for various research, and it justifies the global analysis of food security, human and economic development, poverty, and environmental health [2]. Thus, quantification of water scarcity can play a critical role in locating hotspot regions and help in decision-making for planning and management purposes.

Various approaches have been developed to assess the variability of the blue and green water resources and estimate water scarcity since the introduction of the concept. Schuol et al. [11] and Faramarzi et al. [12] modelled blue-green water availability in Africa and Iran, respectively. Zang et al. [13] assessed the impact of human activities on blue-green water flow in a river basin in China. Veettil & Mishra [4] quantified blue and green water resources of a watershed using a water footprint concept. Abbaspour et al. [14] assessed the impact of climate change on blue and green water resources by using a Canadian Global Coupled Model (CGMC 3.1) in Iran. Also, various modelling pathways have been used by researchers to assess water resources, ranging from global hydrological models (GHMs), like WaterGap 3.0 [15], to catchment-scale hydrological models (CHMs), like the soil and water assessment tool (SWAT) [16]. Faramarzi et al. [12] used SWAT to model blue and green water availability in Iran. Liu et al. [10] used the GIS-based environmental policy integrated climate (GEPIC) model to estimate global water consumption for crop areas. Other models such as CROPWAT [17] and AquaCrop [18] have also been used to quantify water scarcity.

One of the critical issues in water and natural resource management is to satisfy human needs without compromising environmental stability [19]. However, human activities such as excessive withdrawal, overfishing, and water pollution have damaged the ecological balance and impaired freshwater biodiversity [19,20]. Many studies have been conducted to assess global water security [21–23], but most of them overlooked the water requirement for environmental sustainability, also known as the environmental flow requirement (EFR), and only a limited number of studies have included the environment flow component [24–26]. So, there is a need to deliver certain shares of available freshwater for environmental sustainability and evaluate the same using suitable methods.

Canada is one of the most bestowed countries with respect to water availability, with around 7.6% of its total area comprised of water resources [27]. It ranks high in river flow as well, with Canadian rivers discharging nearly 9% of the world's water supply [28]. But this aggregated data

is somewhat deceiving. Being a large landmass, Canada encounters huge spatial and temporal variabilities in climatic conditions, distributions of water resources, and associated water availability, which is considered to be the essence of water scarcity. In this study, we modelled the Grand River watershed (GRW), which is one of the largest watersheds in southwestern Ontario, Canada. The GRW encompasses an area of 6700 km$^2$ and has a population of about 1 million, which is expected to increase beyond 1.5 million by 2051 [29]. In the 1800s during European settlement, land use in some areas was converted to agricultural and urban areas, which led to a 65% loss of wetlands and considerable deforestation as well. Consequently, the river flow became uncontrollable and flashy, causing floods during spring and severe droughts during summer [30]. Since then, many improvements have been made after the formation of the Grand River Conservation Commission (GRCC) in 1932, but no study has been done to analyze water resource variability and water scarcity in the river basin. Furthermore, the GRW has some rapidly growing cities in Ontario, including Kitchener, Waterloo, Cambridge, Guelph, and Brantford. An increase in population and industrialization coupled with climate uncertainty have increased concerns about the ecological impacts and the ability of surface and groundwater to sustain the increasing demand [31]. Therefore, it is important to know how freshwater resources are distributed and used in the watershed to better understand its variability and assess the impact of human activities on watershed hydrology.

Hence, the primary aim of this study is to quantify available water resources in the GRW to estimate water security. This leads to several specific objectives: (a) to build, calibrate, validate, and perform sensitivity as well as uncertainty analyses of a SWAT model of the GRW; (b) to assess spatial and temporal variabilities of blue and green water resources of the GRW; and (c) to assess the water security situation at different zones of the GRW using green and blue water footprints. To our knowledge, this is the first study of this kind in which any basin in Canada, in general and the GRW in particular, has been assessed for its fresh water resources, and a robust analysis of the water security situation has been carried out. Furthermore, we evaluated three different methods for calculating EFR and their sensitivities in quantifying blue water security for the first time. This modelling approach can also be used in other cold climates and snow-dominated regions, which are assumed to be water-abundant but are actually water-scarce. We believe that the availability of a calibrated and validated SWAT model and the know-how of long-term trends of fresh water resources and water security of the GRW will help water resource managers and planners to manage the GRW in a more sustainable way.

## 2. Materials and Methods

### 2.1. Study Area

The Grand River watershed (GRW) is one of the largest watersheds in southwestern Ontario, Canada (Figure 1). The Grand River starts from the headwaters in the Dundalk Highlands and drains into Lake Erie from the outlet at Port Maitland. Along its 310 km-long route it picks up its major tributaries: the Conestoga, the Nith, the Eramosa, and the Speed Rivers [32]. The major areas in the GRW are comprised of agriculture (43%), followed by pastures and range-grasses (26.92%), forests (12%), small fragments of urban areas (9.29%), and wetlands (1.8%) (Figure 2). The weather in the GRW is moderate to cool-temperate (average annual precipitation ranging between 800–900 mm and temperature between 8–10 °C [33]), and the watershed experiences four main seasons including winter, which is cool and dry, and summer, which is hot and humid. If the winter is warm then the watershed will experience moderate spring flow, and if it is cold, then the spring flow could be high enough to cause floods in the downstream areas [34], as evidenced in Brantford in 2018. The Brantford floods were the worst floods that the area had experienced in the last decade, which caused around 4900 residents to move from their homes to safer places [35].

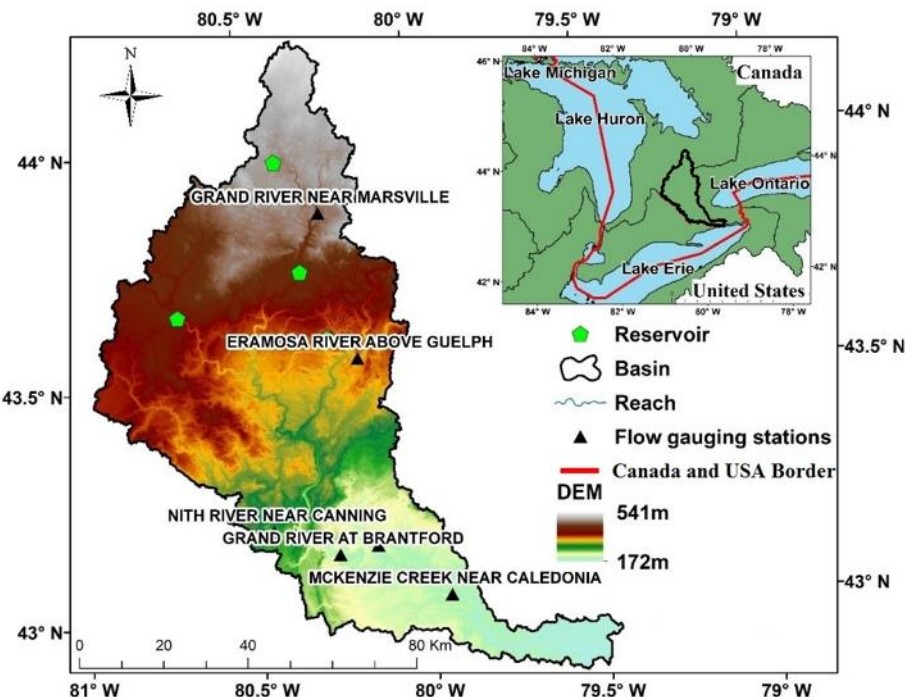

**Figure 1.** The Great Lakes (inset) with the Grand River basin and a digital elevation model (DEM). Also shown are the six gauging stations and the river network derived from the DEM.

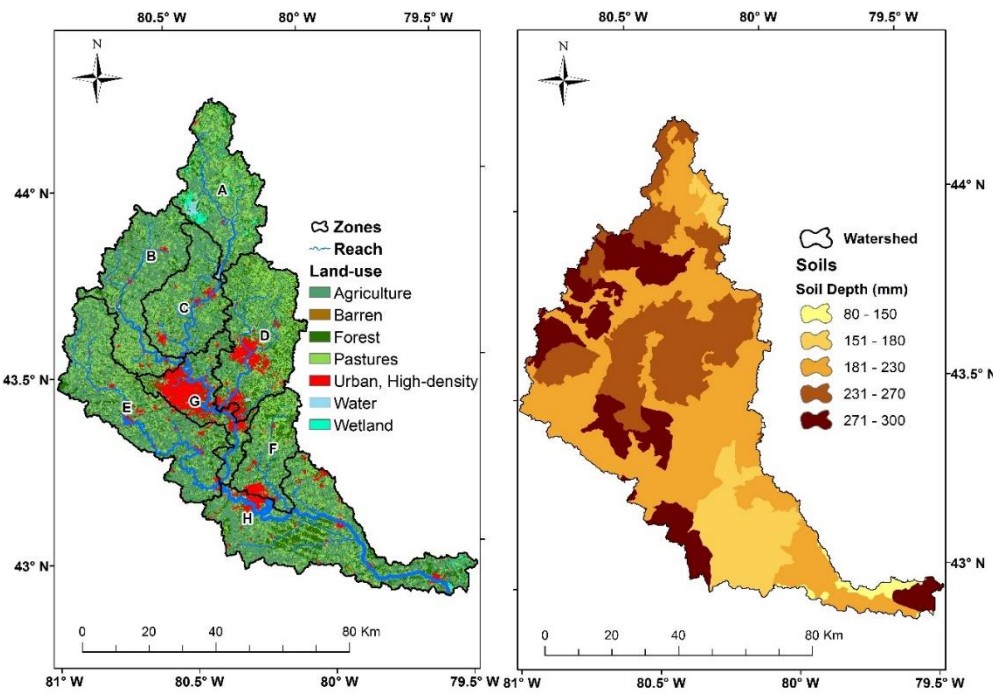

**Figure 2.** Land use and soil depth map of the Great River watershed (GRW).

The precipitation is fairly uniform with short but intense rainfall events in summer and spring, steady rainfall events in autumn, and snowfall in winters. The major areas of the watershed contain the Guelph soil type (14%), followed by Huron (13%), Brantford (12%), and Perth (8%). The elevation of the GRW ranges from 173 m near the outlet at Lake Erie and 535 m near the headwaters. The watershed is divided into eight zones (Figure 3) based on river drainage.

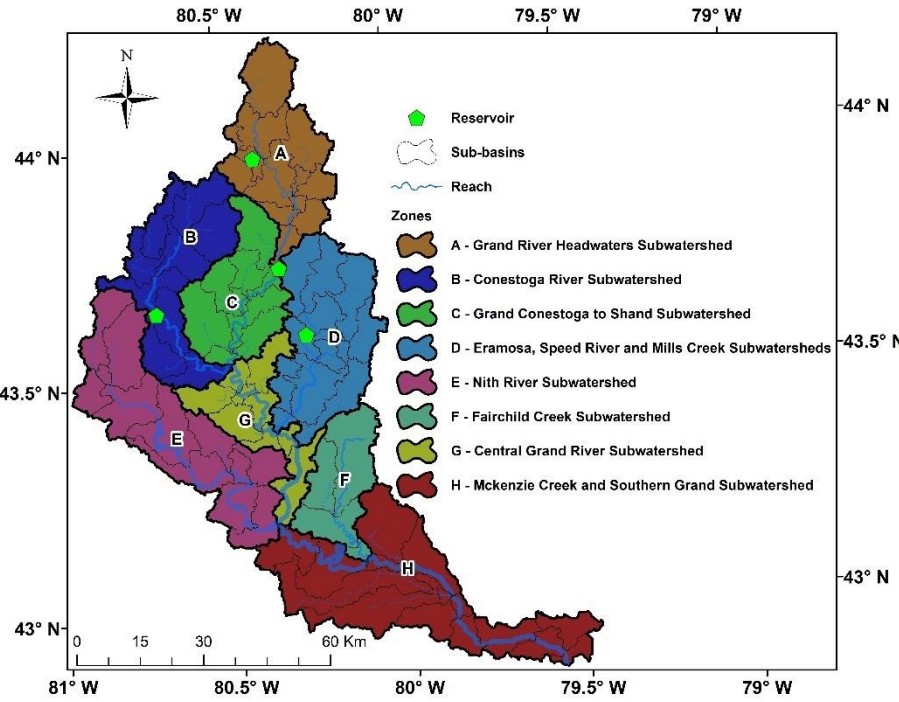

**Figure 3.** Eight different zones in the GRW.

## 2.2. The Soil and Water Assessment Tool (SWAT)

The soil and water assessment tool (SWAT) [16] is a physically based and semi-distributed hydrological model that operates on a daily time-step. The model simulates various eco-hydrological and anthropogenic processes using the water balance equation, and it considers processes such as precipitation, infiltration, percolation, surface runoff, evapotranspiration, and lateral flow. The model has been used widely for simulating snow, standing water, crop growth, and water quality and quantity [36–38]. SWAT simulates the watershed in two phases, land phase and routing phase [39]. The land phase controls the discharge, nutrient, pesticide, and sediment loading from each sub-basin, and the routing phase controls the movement of the water from sub-basins to the main outlet. The watershed is divided into sub-basins, which are further divided into hydrological response units (HRUs) based on similar land use, topography, and soil characteristics. It simulates all the processes at the HRU level, adds them up to get the flow at the sub-basin level, and routes the flow through reaches to the main watershed outlet [16]. Figure 4 shows a flow chart of the SWAT model in assessing water security using blue and green water footprint concepts in the GRW, which is described in detail in the following sections.

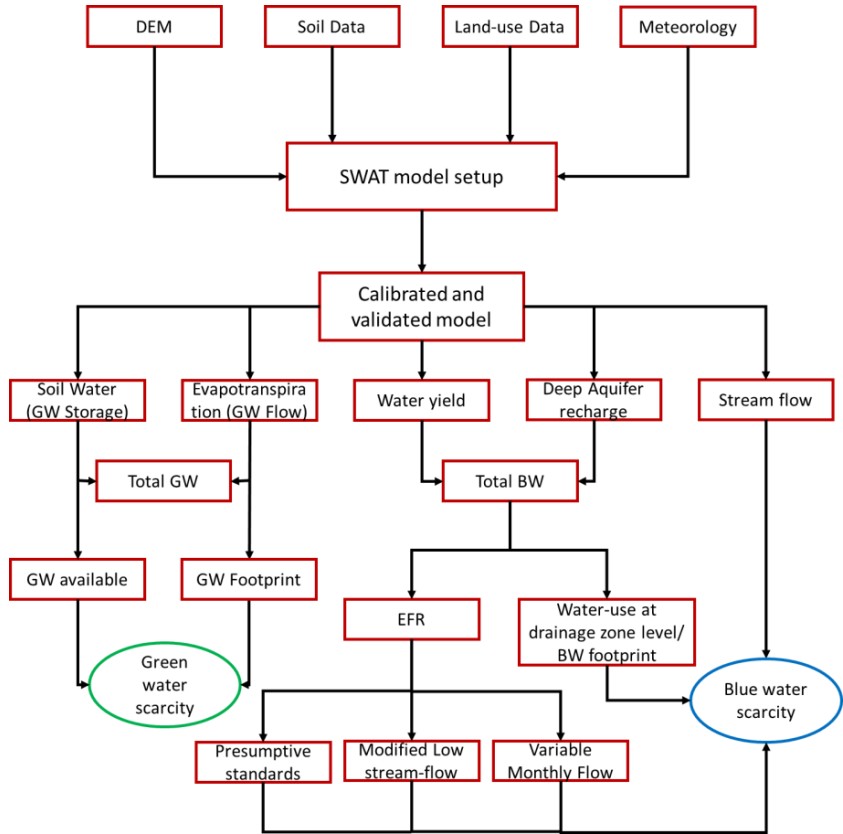

**Figure 4.** Modelling framework using soil and water assessment tool (SWAT) model outputs for water security assessment.

### 2.3. Model Build-Up, Sensitivity Analysis, Calibration and Validation, and Uncertainty Analysis

A SWAT model requires various spatial, land, and crop management hydro-meteorological datasets. Table 1 briefly summarizes the data used, their sources, and their spatial and temporal resolutions.

**Table 1.** Input data used for the SWAT model of the GRW.

| Data | Source | Resolution |
|---|---|---|
| Topography (Digital Elevation Model—DEM) | Ontario Ministry of Natural Resources and Forestry (OMAFRA) | 30 × 30 m |
| Soil | Soil Landscapes of Canada (SLC), version 3.2 | 1:1 million |
| Land use | Agriculture and Agri-Food, Canada (AAFC) | 30 × 30 m |
| Crop management | Agriculture and Agri-Food, Canada | 30 × 30 m |
| Tile drainage | Ontario Ministry of Agriculture, Food and Rural Affairs (OMAFRA) | 30 × 30 m |
| Precipitation | Natural Resources Canada (NRCAN, 2018) | Daily (mm) |
| Maximum and Minimum Temperature | Natural Resources Canada (NRCAN, 2018) | Daily (°C) |

A digital elevation model (DEM) of 30 × 30 m spatial resolution obtained from the Ontario Ministry of Natural Resources and Forestry [40] was used to discretize the watershed in smaller sub-basins. Moreover, it was used to obtain a slope map, which was further divided into three slope classes with breaks at 2% and 4%. A soil map of resolution 1:1 million was obtained from Soil Landscapes of Canada (SLC), version 3.2. Similarly, a land-use map was prepared using crop data layers (CDL) for the years 2014, 2015, and 2016, obtained from Agriculture and Agri-Food, Canada, using

the procedure documented by Srinivasan et al. [41,42]. Furthermore, this spatial layer was merged with a tile drainage layer obtained from the Ontario Ministry of Agriculture, Food and Rural Affairs (OMAFRA) to derive a land use/land cover map of 30 m resolution. Thresholds of 5% each for land use, soil, and slope were used to delineate HRUs, which resulted into a total of 2679 HRUs distributed over 85 sub-basins. The characteristics (e.g., location, operation period, and physical and structural characteristics) of four reservoirs, the Guelph dam, Luther dam, Conestoga dam, and Shand dam, were implemented in the model as they would affect the streamflow dynamics of the GRW. We used the simulated target release option to mimic operations such as storing spring floods and releasing them in summer months. In addition, minimum monthly outflows were also allocated based on streamflow data measured downstream from the reservoir so that the real-world reservoir operations were reasonably represented. Long-term (1950–2015) weather data of resolution $10 \times 10$ km at daily time intervals was obtained from Natural Resources Canada [33]. The Soil Conservation Services (SCS) curve number method was used for runoff estimation, while the Hargreaves method [43] was used for estimating evapotranspiration.

Agricultural crop growth is another essential factor that affects the hydrological cycle. Moreover, the crop/land management operations involved, such as irrigation and fertilization, affect the hydrology and water quality as well. Therefore, a three-year crop rotation of corn, soybean, and winter-wheat, as typically used in Ontario, and their common management operations such as planting, fertilizer application, and harvesting, etc., were taken from Liu et al. [36] and implemented in the model. While a robust calibration of crop yield was not the focus of the study, we did tun crop yield-related parameters to match the simulated crop yield with those obtained from OMAFRA for a proper and accurate hydrological representation of the watershed.

Multi-site calibration and validation techniques, as recommended by Daggupati et al. [44], were used considering monthly stream flows at six different flow-gauging stations (Figure 1). As such, the total time period (1950–2015) was divided into a warm-up period (1950–1952), validation periods (1970–1981 and 2008–2015), and a calibration (1982–2007) period. As streamflow data for the period 1953–1969 were not available, calibration and validation were done in the period 1970–2015. The purpose of choosing two validation periods was to include both dry as well as wet years, which was validated using a standard normal variate (z-score) of annual precipitation. To facilitate multi-site calibration and validation as well as sensitivity and uncertainty analysis, we used the SWAT-CUP [45] and its sequential fitting program (SUFI-2) algorithm.

In order to evaluate the performance of the model, we followed the recommendations of Moriasi et al. [46]. As such, a coefficient of determination ($R^2$), Nash–Sutcliffe Efficiency (NSE), and percentage of bias (PBIAS) were selected, and one of the four qualitative ratings (very good, good, satisfactory, and unsatisfactory) was assigned based on the range of the value of the chosen goodness-of-fit statistics. Two indices were used for uncertainty analysis: the p-factor, which is the percentage of measured data bracketed by the 95% predictive uncertainty (PPU) band, and r-factor, which is the measure of average width of the band [45]. The p-factor has an optimal value of 1, which means 100% of the observed data is bracketed in the uncertainty band, while for the r-factor, the lower the value the more certain the model results. Additionally, various time-series plots and scatter plots were used for the analysis.

### 2.4. Spatial and Temporal Quantification of Blue Water and Green Water Resources

As already depicted, the GRW was divided into eight spatial zones based on river drainages to the Grand River (Figure 3) and was used to quantify freshwater resources and water security analysis. Blue water constituted the total water yield from the watershed and the deep aquifer recharge [8]. The total water yield was the amount of water leaving the HRU flowing into the main channel, and groundwater storage was the water moving downwards from the sub-surface to a deep aquifer [8]. Both water yield and deep aquifer recharge can be found from SWAT HRU output tables. Green water resources included green water flow (i.e., water flowing through evapotranspiration) and green

water storage (i.e., water stored in the soil as soil moisture) [11]. Evapotranspiration and soil moisture content can also be obtained from SWAT output tables at an HRU spatial scale. In addition to spatial analysis, the freshwater resources (obtained from SWAT output files) were also analyzed for their temporal variations using Sen's slope method [47] and z-variate analyses for the period 1953–2015. In the Sen's slope method, the trend was deduced by calculating the median of all pairwise slopes of alternate values in the dataset. Moreover, monthly and seasonal variations of blue and green water resources were analyzed. Uncertainty in these freshwater resource components were represented with a 95% PPU, as calculated using the SWAT-CUP.

*2.5. Quantification of Blue Water Scarcity*

Blue water scarcity was computed as the ratio of consumptive use of blue water and the total blue water available. The blue water use, as discussed earlier, was obtained from the report Water Use Inventory Report for the Grand River Watershed [48]. Water use was divided into agricultural, municipal, recreational, dewatering, industrial, and remediation purposes. The major water use sector was municipal (60.83%), followed by water needed for dewatering (6.07%), agricultural (4.47%), and agricultural livestock (4.41%). The blue water availability (Equation (1)) is the finite amount of water which can be abstracted for human consumptive use without sacrificing environmental stability [4]. It can be computed as the stream flow minus the EFR [9], which is the amount of water required to sustain the ecosystem. In the GRW, a significant amount of water demand is also met by groundwater in addition to surface water [48], accordingly, ground water storage/deep aquifer recharge (DAR) was also considered while calculating blue water availability.

$$\text{BW(available)}_{(x,t)} = Q_{(x,t)} - \text{EFR}_{(x,t)} + \text{DAR}_{(x,\,t)}, \tag{1}$$

where $\text{EFR}_{(x,t)}$ is the environment flow requirement for drainage area 'x' at time 't', $\text{DAR}_{(x,t)}$ is the deep aquifer recharge at the same drainage area 'x' and time 't', and $Q_{(x,t)}$ is the corresponding stream flow in $m^3/s$.

2.5.1. Different Methods to Calculate the Environment Flow Requirement (EFR)

To assess the feasibility of an area to sustain and increase in agricultural production or any other water-intensive job, we must acknowledge the fact that the environment also uses water, and that to maintain an eco-hydrological balance, limits must be set to withdraw water in time and space [19]. In this study, three different hydrological-based methodologies were used to compute EFR, which will be described in following sections.

Presumptive Standards Method

This approach was proposed by Richter et al. [49] and Hoekstra et al. [25], and it states that only 20% of the flow is available for use, and the remaining 80% is the EFR. However, it does not mean that 80% of the flow is unavailable for use. In reality, 100% of the flow is available for use, but no more than 20% of the flow should be depleted by consumptive water use. It has been used in various water security studies such as [2,4,8,9]. This method is a precautionary approach. It is generally used where site-specific estimation of EFR cannot be made and detailed local studies cannot be completed in the near-term [49].

$$\text{EFR}_{(x,t)} = 0.8 Q_{\text{mean}(x,t)}, \tag{2}$$

where $\text{EFR}_{(x,t)}$ is the environment flow requirement for drainage area 'x' at time 't', and $Q_{\text{mean}(x,t)}$ is the long-term monthly mean stream flow in $m^3/s$.

Modified Low Streamflow Method ($Q_{7,10}$)

The $Q_{7,10}$ is the average annual seven-day minimum flow that is expected to be exceeded in nine out of ten years, or it is the 10th percentile of the distribution of seven-day monthly minimum flows [50].

The original low streamflow method does not consider intra-annual variability. As such, for some months the EFR could be so high that only a small amount of water was left for use, which in turn could increase blue water scarcity. Therefore, the method was modified by estimating $Q_{7,10}$ values at monthly scales to consider intra-annual variability, and different EFR values were computed for different months. The data set was adjusted close to normality by using the log-normal transformation [51].

Variable Monthly Flow (VMF) Method

The variable monthly flow (VMF) method is a parametric method of EFR estimation, and was proposed by Pastor et al. [19]. This method considers the natural variability of the flow by considering EFR at a seasonal level. It classifies the flow into three classes:

- Low flow (MMF ≤ 40% of the mean annual flow (MAF));
- Intermediate flow (40% of MAF < MMF < 80% of MAF);
- High flow (MMF > 80% MAF);

where, MMF stands for mean monthly flow for a particular month.

Different criteria are used for each class to estimate the EFR, as listed below:

- For low flow:

$$EFR = 0.6 \text{ MMF}; \tag{3}$$

- Intermediate flow:

$$EFR = 0.45 \text{ MMF}; \tag{4}$$

- High flow:

$$EFR = 0.3 \text{ MMF}. \tag{5}$$

Once the blue water scarcity was estimated for each EFR method, four qualitative ratings were assigned the following [2]:

- Low blue water scarcity, with a scarcity value less than 1
- Moderate blue water scarcity, with a scarcity value between 1 and 1.5
- Significant blue water scarcity, with a scarcity value between 1.5 and 2
- Severe blue water scarcity, with a scarcity value more than 2

### 2.6. Freshwater Provision Indicator

The freshwater provision indicator ($FWPI_{quantitative}$) is used to compare the effects of the three EFR methods described above, and it was used in this study. This indicator was proposed by [52]. The $FWPI_{quantitative}$ describes the freshwater provision services based on the quantity of water provided. Its value is based on the natural reasons, such as drought, that EFR levels might be violated, and it measures the risk in terms of the frequency of average monthly flows that are less than the EFR. A value equal to one denotes that EFR is met throughout the time period, and a value less than one means EFR is not met.

$$FWPI_{quantitative} = \frac{Q_{mean(x,t)}/EFR_{(x,t)}}{(Q_{mean(x,t)}/EFR_{(x,t)}) + (qne_t/n_t)}, \tag{6}$$

where $FWPI_{quantitative}$ stands for the freshwater provision indicator for the zone 'x' and time 't', $qne_t$ stands for the number of months for which the $FWPI_{quantitative}$ values are less than EFR, and $n_t$ stands for total number of months considered.

### 2.7. Quantification of Green Water Scarcity

In this study, green water scarcity was used to evaluate green water security in the watershed. Green water scarcity was calculated as the ratio of green water (GW) footprint (i.e., evapotranspiration) to the amount of water available for use (i.e., soil water content) [9]. As already depicted, we chose the

Hargreaves method [43] while running the SWAT for simulating evapotranspiration (ET), and SWAT simulated the ET at HRU spatial scale. We used initial soil water content ($SW_{in}$) as available green water, which was obtained from SWAT HRU output files. Available green water is the amount of water available to sustain plant growth and to fulfil the evapotranspiration need/consumptive need of the plant system. It is calculated as total water present in the root zone minus water content at wilting point [4,8], which is also the minimum water required for sustainable growth of the plant. Hence, green water scarcity can be estimated with (Equation (7)):

$$GW_{scarcity} = \frac{GW_{footprint\ (x,t)}}{GW_{available\ (x,t)}}, \tag{7}$$

where $GW_{footprint\ (x,t)}$ is the amount of green water consumed or evapotranspired in zone 'x' during the time 't', and $GW_{available\ (x,t)}$ is the initial soil water content present in zone 'x' for time 't'.

## 3. Results and Discussion

### 3.1. Performance Evaluation of SWAT Model Results

A total of 17 SWAT parameters (Table 2) were subjected to calibration based on the sensitivity analysis performed using the SWAT-CUP global sensitivity analysis. While the calibration was carried out using a wider range of parameter values, a final parameter range was presented (Table 2). We found the SCS curve number for moisture condition II (CN2) to be the most sensitive parameter, followed by the baseflow recession constant (ALPHA_BF). The ground water delay time (GW_DELAY) completed the top three list. Two snow-related parameters, snowfall temperature (SFTMP) and snowmelt temperature (SMTMP), were also in the top five list, which was indeed expected as the GRW is in a cold climate region. The sensitiveness of the parameters was fairly comparable to that reported in other studies conducted in similar regions [37,53,54].

**Table 2.** Sensitive parameters obtained from SWAT-CUP used for model development.

| Sensitive Parameters | Description | Default | Minimum | Maximum |
|---|---|---|---|---|
| r_CN2.mgt | SCS curve number for moisture condition II (-) | HRU * | −10% | 15% |
| v_ALPHA_BF.gw | Baseflow recession constant (-) | 0.048 | 0.4 | 0.7 |
| v_GW_DELAY.gw | Groundwater delay time (days) | 31 | 10 | 100 |
| v_SFTMP.bsn | Snowfall temperature (°C) | 1 | −3 | 1 |
| v_SMTMP.bsn | Snowmelt base temperature (°C) | 0.5 | 0 | 5 |
| v_ESCO.bsn | Soil evaporation compensation factor (-) | 0.95 | 0.9 | 1 |
| v_EPCO.bsn | Soil uptake compensation factor (-) | 1 | 0.7 | 1 |
| v_GWQMN.gw | Threshold depth of water in shallow aquifer required for return flow to occur (mm) | 1000 | 500 | 1500 |
| v_GW_REVAP.gw | Groundwater revap. Coefficient (-) | 0.02 | 0.1 | 0.2 |
| v_REVAPMN.gw | Threshold depth of water in shallow aquifer for revap to deep aquifer to occur (mm) | 750 | 650 | 850 |
| r_SOL_K.sol | Soil hydraulic conductivity (mm/h) | soil type ** | −10% | 10% |
| r_SOL_AWC.sol | Available water capacity of soil layer (mm/mm) | soil type ** | −10% | 10% |
| v_CH_N2.rte | Manning's n value for the main channel (-) | 0.14 | 0.03 | 0.2 |
| v_CH_K2.rte | Effective hydraulic conductivity in main channel alluvium (mm/h) | 0 | 10 | 100 |
| v_ALPHA_BNK.rte | Baseflow alpha factor for bank storage (days) | 0 | 0.2 | 0.6 |
| v_TIMP.bsn | Snow pack temperature lag factor (-) | 1 | 0.5 | 1 |
| v_SMFMX.bsn | Melt factor of snow on June 21 (mm H2O/°C-day) | 4.5 | 2 | 6 |

\* depends on the HRU,hydrological response unit (HRU), \*\* depends on the soil type

r_: relative change with the value with respect to the original (default) value

v_: replaced by the given value

Table 3 shows the goodness-of-fit statistical values for calibration and validation as well as sensitivity analysis of streamflow results at six gauging stations. Figure 5 shows the graphical comparison of observed and simulated stream flows with a 95% uncertainty band. Graphical plots show that the model, in general, had represented observed trends, and most of the observations fell within the 95% uncertainty band; the values of chosen goodness-of-fit statistics and both p- and r-factors indeed reflected that. For all six stations, the uncertainty band encapsulated at least 81% of observations (as the p-factor ranged between 0.81–0.93 in the calibration period). Moreover, the r-factor, which indicated the thickness of the band, was also less than the generally accepted value of 1.5 [3] (0.81 to 1.24, case in calibration period). It should be noted that a higher p-factor could have been obtained by increasing the r-factor, but a compromise had to be made to not introduce further uncertainty in the model results [45]. The qualitative ratings, calculated based on the recommendation of Moriasi et al. [46], indicated a range of accuracy (unsatisfactory to very good), varying as per the goodness-of-fit statistics and gauging stations. In general, the model simulated streamflow dynamics of downstream stations with better accuracy as compared to those at tributaries. A slightly lower accuracy was observed for the validation period when compared to those in the calibration period, which was indeed expected as the model tends to underperform in the validation period. The results of the model were also found to be comparable with the result-statistics calculated in other cold regions in Canada [37,38,53–56].

**Table 3.** Goodness-of-fit statistics obtained from calibration (1982–2007) and validation (1970–1981; 2007–2015) at six gauging stations.

| Flow Station | Calibration Results | | | | | Validation Results | | | | |
|---|---|---|---|---|---|---|---|---|---|---|
| | p-Factor | r-Factor | Nash–Sutcliffe Efficiency (NSE) | $R^2$ | Percent Bias (PBIAS) (%) | p-Factor | r-Factor | NSE | $R^2$ | PBIAS (%) |
| Grand at Marshville | 0.83 | 0.81 | 0.82 (G) | 0.83 (G) | 8.60 (G) | 0.71 | 0.67 | 0.83 (G) | 0.86 (V) | 19.8 (U) |
| Eramosa at Guelph | 0.85 | 1.33 | 0.66 (U) | 0.68 (U) | 0.40 (V) | 0.77 | 1.2 | 0.69 (U) | 0.79 (S) | 9.30 (G) |
| Grand at Brantford | 0.81 | 0.99 | 0.83 (G) | 0.85 (V) | 11.4 (S) | 0.91 | 1.14 | 0.8 (G) | 0.86 (V) | 18.8 (U) |
| McKenzie at Caledonia | 0.84 | 0.99 | 0.74 (S) | 0.74 (S) | 1.60 (V) | 0.88 | 0.96 | 0.57 (U) | 0.62 (U) | 18.1 (U) |
| Fairchild at Brantford | 0.87 | 0.99 | 0.76 (S) | 0.76 (S) | −0.80 (V) | 0.76 | 0.88 | 0.63 (U) | 0.67 (U) | 3.40 (G) |
| Nith at Canning | 0.93 | 1.24 | 0.87 (V) | 0.87 (V) | 5.90 (G) | 0.83 | 0.98 | 0.81 (G) | 0.82 (G) | 2.30 (V) |

V: very good; G: Good; S: satisfactory; U: unsatisfactory, as per recommendations of Moriasi et al. [46].

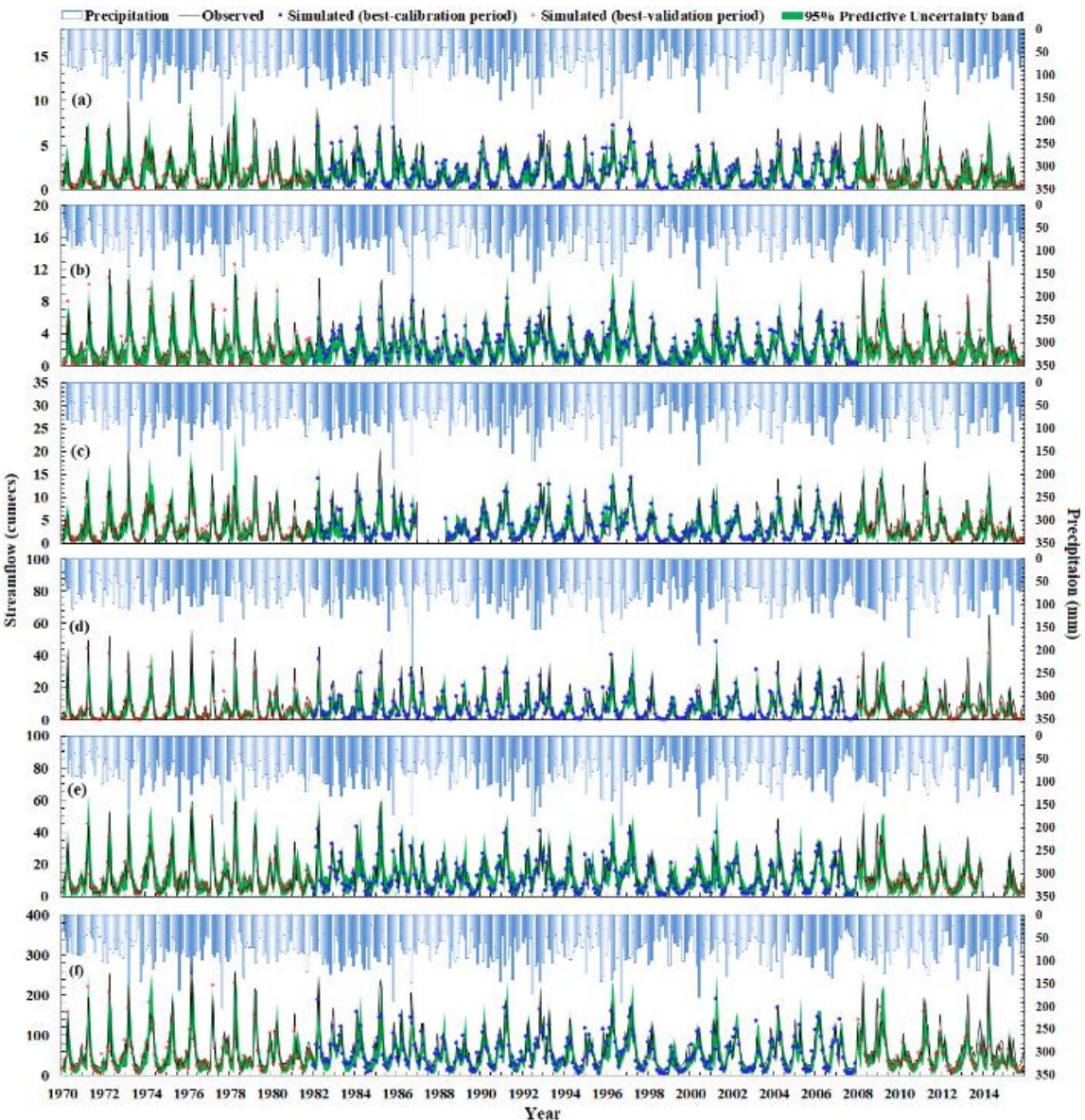

**Figure 5.** Time series plot showing the observed and simulated monthly flow for six gauging stations: (**a**) Grand River at Marshville (02GA014), (**b**) Eramosa and Speed River at Guelph (02GA029), (**c**) Nith River at Canning (02GA010), (**d**) Fairchild Creek near Brantford (02GB007), (**e**) Grand River at Brantford (02GB001), and (**f**) Mckenzie Creek near Caledonia (02GB010).

### 3.2. Spatiotemporal Variability of the Fresh Water Resources

The blue water yield showed marked spatial variability and, as expected, a high correlation with variability of the annual average precipitation (Figure 6). Blue water varied from 222 mm in the southeastern part of the watershed to 430 mm in the western part of the watershed. Sub-basins located at the western parts of the basin (upstream parts of the Conestoga and Nith sub-watersheds) had the highest blue water yields. Particularly, a sub-basin in the western part of the watershed showed an extremely high value of blue water (430 mm) as compared to the watershed average of 301 mm, which was mainly because of the presence of Brookston clay soil that has poor drainage properties. This resulted in limited infiltration, and as such, a higher blue water flow. Amongst different zones, the Fairchild Creek sub-watershed (Zone F) contributed the least, which can be directly related to the lower amount of precipitation it received when compared to other parts of the watershed.

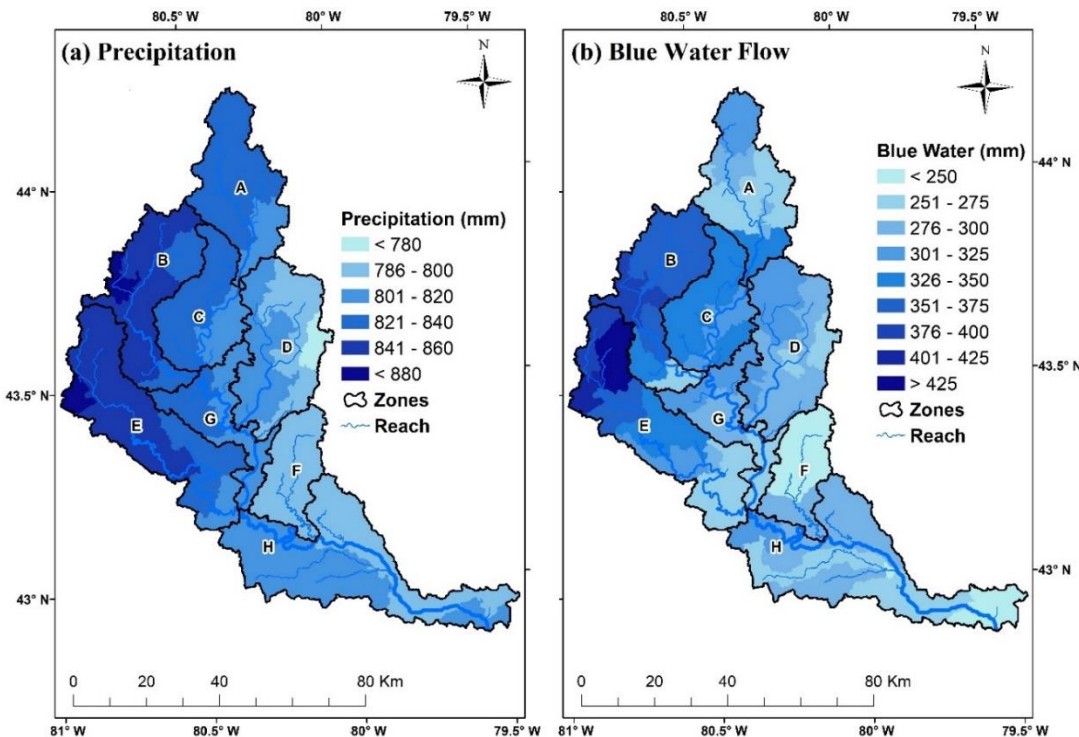

**Figure 6.** Spatial distribution of average annual precipitation and blue water flow.

The inter-annual variability based on a long-term (63 years: 1953–2015) assessment of the blue water showed an interesting trend (Figures 7 and 8). While blue water yield showed an increasing trend using Sen's slope analysis, a kind of cyclic pattern was evident, with increasing values in some years (positive z-value; Figure 8 in years 1975–1980) and decreasing values in other years (negative z-values; Figure 8 in years 1960–1965). Despite a higher rate of temperature increase, the blue water resources increased steadily, which can be attributed mainly to an increase in precipitation. Hence, inter-annual variability in precipitation, as expected, exerted higher control on the inter-annual variability of blue water than that of temperature. As evident in the z-score plot, an increase or decrease in precipitation had similar effects on different zones (A to H) of the watershed, as indicated by the same colored blue water yield z-score values, highlighting consistent responses to precipitation fluctuations among the considered eight zones of the GRW.

Figure 7 also shows the uncertainty band (95% PPU). It should be noted that a thicker band (with band thickness varying from 200 mm to over 400 mm) was obtained for blue water resources when compared to the same for green water resources, which implied higher variation in the blue water resources and, hence, higher uncertainty. This higher uncertainty was evident from the fact that the blue water was sensitive to more parameters as compared to other freshwater components [11] such as green water flow, which is primarily dependent on diurnal temperature fluctuations. A small change in precipitation can more quickly affect blue water flow than green water flow and storage. As for the green water, it was evident that both green water flow and green water storage were associated with temperature (Figure 9). The green water flow varied from 425 mm/year in upstream sub-basins, where the temperature was low, to 575 mm/year in the downstream temperature where the mean annual temperature was higher (more than 9 °C). The spatial analysis of the green water flow (Figure 9) showed lesser variations as compared to the blue water flow. The reason for this is that soil had a limited amount of water storage to compensate for green water flow needs of the vegetation [11]. It should be noted that in the northern parts of the watershed, there was a small sub-basin with extremely high green water flow (Figure 9), although it was in a lower temperature region. The reason for this was higher evaporation occurring from this sub-basin, as most of its area contained water and wetlands. The Luther dam is situated in this small sub-basin, which caused a higher green water flow.

Similarly, the green water storage or the available soil water also showed marked spatial variability. It is indeed true that green water storage is dependent on various soil properties such as depth of the soil layer, hydraulic conductivity, porosity, etc. In our study, the major influencing factor was the soil depth (Figures 2 and 9). Furthermore, the spatial variability in green water storage is also related to variability in precipitation. However, an opposite pattern was observed, especially for the western parts of the GRW. In these areas, while precipitation was the highest, the green water storage was not.

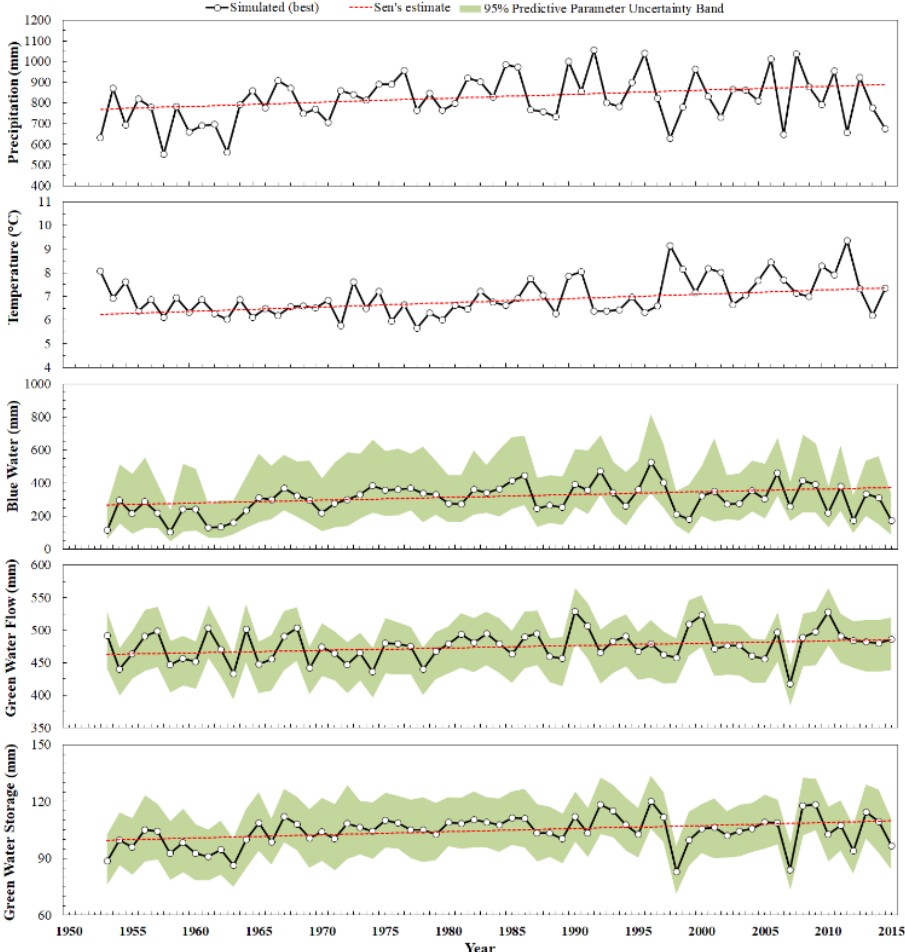

**Figure 7.** Trend analysis using Sen's Slope method of average annual precipitation, temperature, blue water and green water flow, and green water storage.

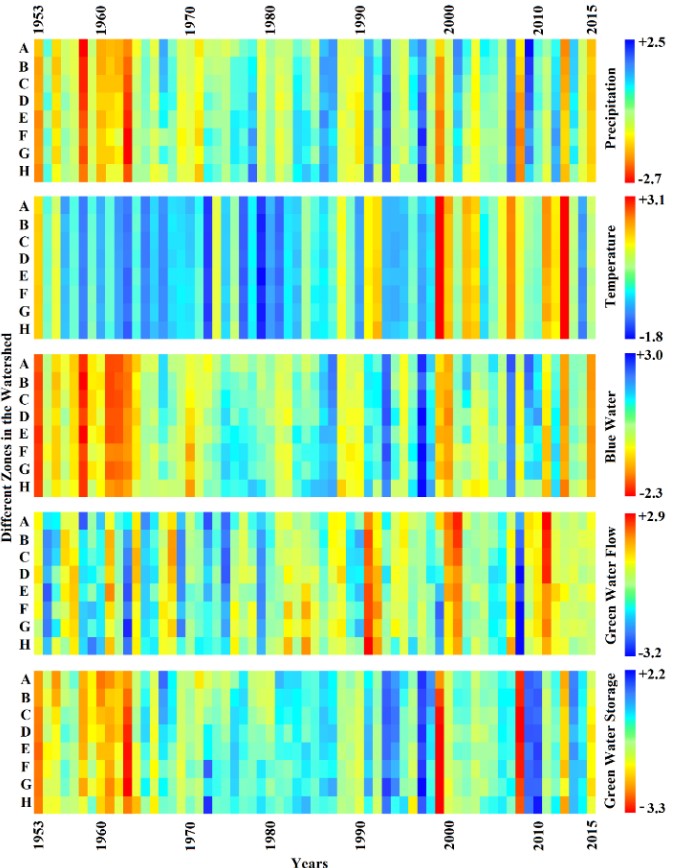

**Figure 8.** Z-variate analysis of average annual precipitation, temperature, blue water and green water flow, and green water storage at the zonal level.

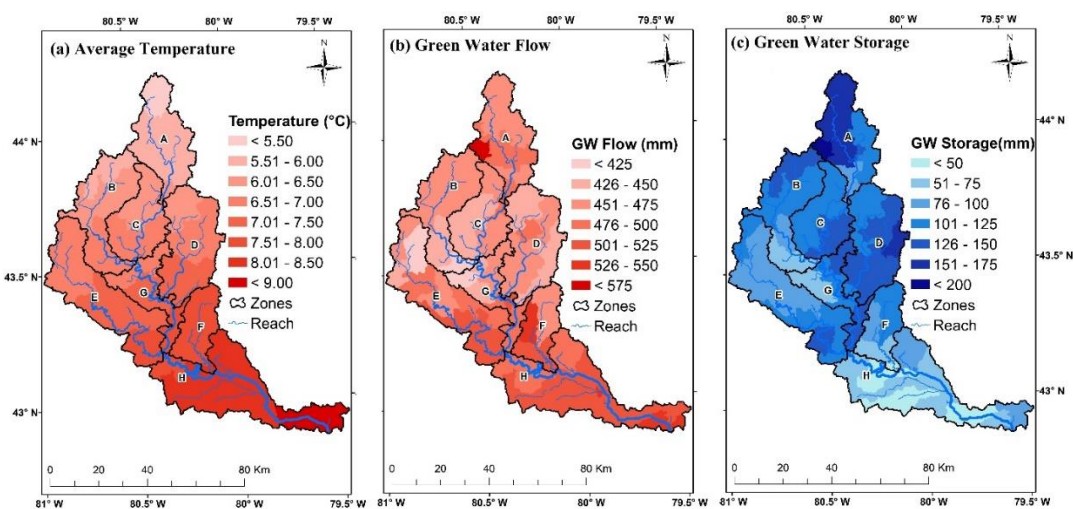

**Figure 9.** Spatial distribution of average annual temperature, green water flow, and green water storage.

The inter-annual temporal and uncertainty analysis for a period of 63 years (1953–2015) showed an increasing trend of both green water flow and storage with a narrow uncertainty band (Figures 7 and 8). As both precipitation and temperature trends showed increasing trends (Figure 7), and their effects on green water storage were compensating, it is evident that the green water storage trends have milder slopes than that of precipitation and temperature. In general, a higher precipitation would allow more infiltration and more green water storage. At the same time, a higher temperature would

mean a higher rate of green water flow. Moreover, a consistent pattern was observed amongst the different zones of the GRW, as indicated by the same color pattern (vertical axis) in the z-score plot for a particular year. As we proceed on the horizontal axis in the z-score plot, temporal variations were evident such as higher green water storage in the years 1980–1985 and lower green water storage in the years 1960–1965. An interesting trend in the z-score plot of green water flow was observed in the years 1960–1965. While green water flow showed a mixed pattern in that period, the pattern of green water storage was more similar to that of precipitation, despite an opposite temperature pattern. This indicates that variability in precipitation can explain variability in green water storage.

The intra-annual variability of the freshwater components was also analyzed using monthly average plots at the zonal level (Supplementary Figure S1a,b). Precipitation was found to be more or less uniform throughout the year, but blue water was the highest in March and April. The peaks were the result of the snowmelt contribution to the total yield in the GRW. Moreover, there was a little rise in blue water in November and December, which might be because of slightly more precipitation in those months. Green water flow was found to be highest from May to July, because of higher temperature and more evapotranspiration in these months. Green water storage also decreased during summer months from May to July, owing to more evapotranspiration in those months.

### 3.3. Blue Water Security Status Using Different EFR Methods

Results showed that each of the EFR methods attributed different amounts of blue water for human consumptive use. The modified $Q_{7,10}$ method attributed 13% ± 5% to the EFR requirement while the VMF method raised the contribution to 35% ± 2% (Supplementary Table S1). The presumptive standards method, on the other hand, was the most restrictive and conservative, which attributed 80% of the discharge to the EFR. As such, it was evident that the modified $Q_{7,10}$ method was the least restrictive and conservative, while the opposite was true for the presumptive standards method. The difference in the EFRs was more prominent in the beginning of summer (June), as compared to the end of summer (August) and beginning of fall (September), because variation in the streamflow was the least during the latter months.

The FWPI (quantitative) was also calculated to compare the EFR methods amongst themselves. As already stated, the FWPI estimates the frequency of streamflow that is less than the EFR. It was found that the average annual FWPI for the period of 63 years and for the whole watershed was 0.26 ± 0.04 using the modified $Q_{7,10}$ method, 0.59 ± 0.14 using the VMF method, and 0.93 ± 0.03 when using the presumptive standards method for EFR calculation. The FWPI values helped to build a restrictiveness scale, with the presumptive standards method being the most restrictive and the modified $Q_{7,10}$ method being the least. It also can help to estimate the natural reasons, like droughts, for violation of the EFR [8]. This analysis can help to select any EFR method based on conservation requirements.

The annual average blue water scarcity (Figure 10) clearly demonstrates the marked variability across different zones. Furthermore, use of different methods while calculating the EFR resulted in different blue water scarcity values. Since modified $Q_{7,10}$ and VMF methods showed lower restrictions in terms of the EFR, it was observed that blue water scarcity values were less than 0.6 (low blue water scarcity) for all the zones. Alternatively, the presumptive standards method showed decreased blue water availability and gave higher values for blue water scarcity, some as high as 1.75 (significant blue water scarcity). Precisely, the Eramosa and Speed river sub-watershed (Zone D) was found to have the most blue water scarcity using all the EFR methods, and it had low green water availability as well (refer next section), further highlighting the effect of urban development in this zone. Water use or the blue water footprint was very high for the Central Grand sub-watershed (Zone G) and Nith River sub-watershed (Zone E) because of the presence of urban centers (Kitchener and Waterloo, Paris and New Hamburg, respectively) in these zones. However, The higher blue water availability under all EFR scenarios compensated for the higher water use and, hence, resulted in lower blue water scarcity.

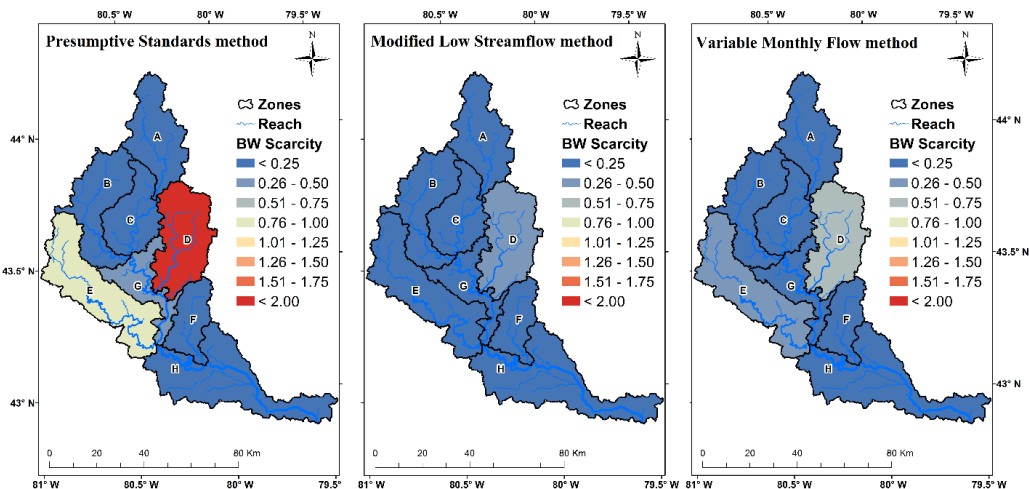

**Figure 10.** Spatial distribution of blue water scarcity (using all three methods for environmental flow requirement (EFR) calculations) for the year 2008.

In the monthly temporal scale, even more significant variability in blue water scarcity existed, depending on the method used for the calculation of the EFR, different zones, and months (Figure 11). For instance, the blue water scarcity was highest in the summer months (May to August), and was consistent in all zones. This was due to higher water use in these months. Furthermore, another peak in blue water scarcity was observed in the month of October for all the zones when using all the EFR methods, on account of lower blue water availability. The blue water scarcity varied between 0.004 (low) and 0.973 (low) for the modified $Q_{7,10}$ method, between 0.005 (low) and 1.31 (moderate) for the VMF method, and 0.017 (low) and 3.27 (severe) blue water scarcity for the presumptive standards.

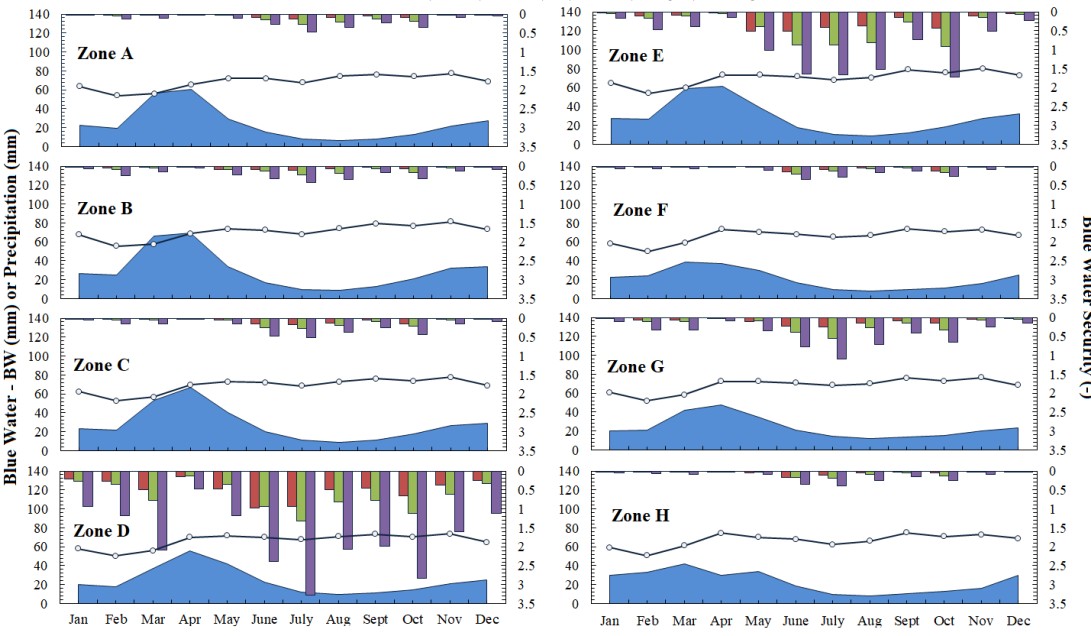

**Figure 11.** Monthly variation of blue water scarcity and its indicators at the zonal level. The graphs show the monthly variation of blue water and precipitation on the primary axis and the blue water scarcity for the year 2008, using all three EFR calculation methods, on the secondary axis.

### 3.4. Green Water Security Status

As observed for the blue water scarcity, the green water scarcity plots (Figure 12 for annual, Figure 13 for monthly time scales) also showed marked spatial and temporal variabilities. The average annual green water scarcity for the whole watershed was found to be 58.82%. Higher values of green water scarcity were seen in the southern Grand and Mckenzie Creek sub-watershed (Zone H) on account of its higher temperature (Figure 9), more agricultural areas, and more pastures resulting in a higher green water flow. Lower green water scarcity was observed in the Eramosa and Speed river sub-watershed (Zone-D) because of its relatively low temperature and higher elevation; hence, there was less green water flow and more green water storage (Figure 9). Even in zones with higher green water flow and lower green water storage, the green water scarcity at an annual level remained less than 1.1 (110%), indicating that green water shortage was not a serious problem in the GRW.

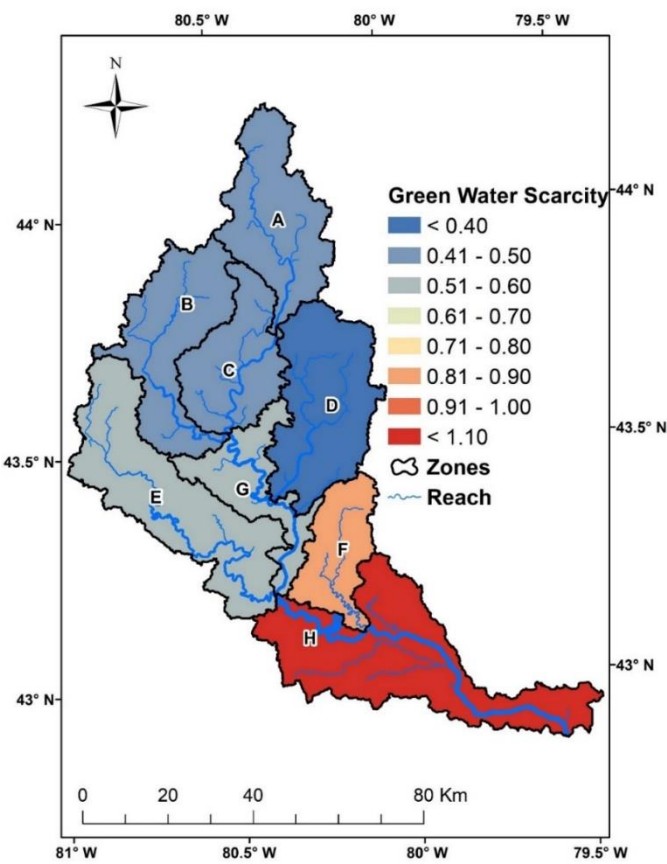

**Figure 12.** Spatial distribution of the average annual green water scarcity.

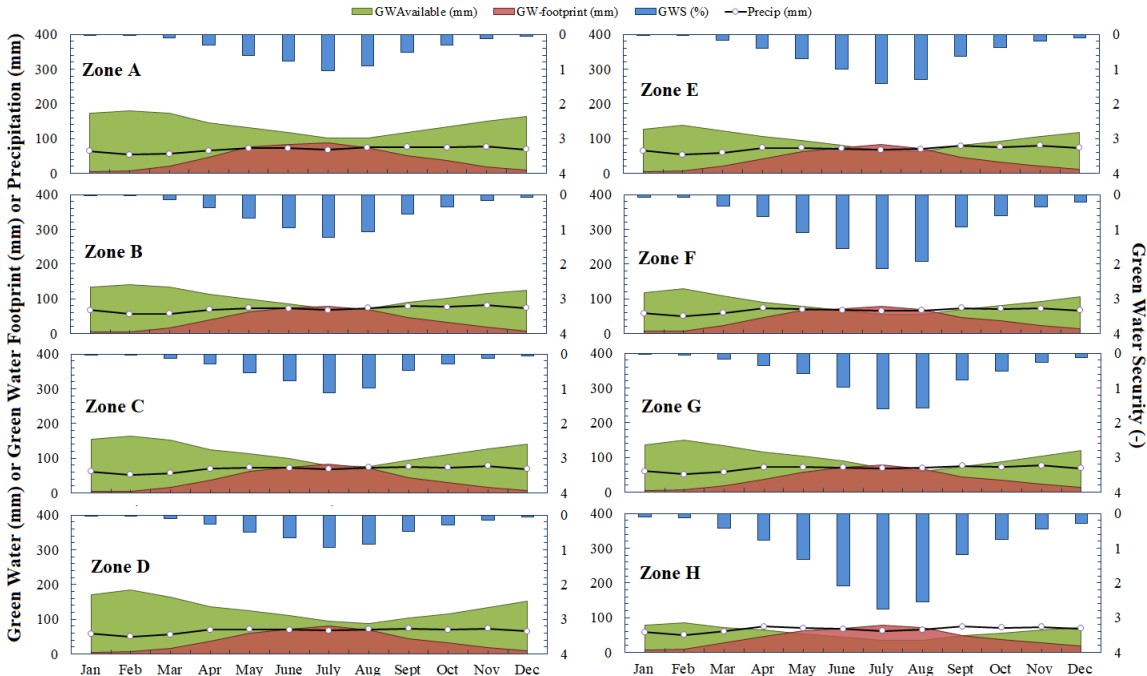

**Figure 13.** Monthly variation of green water scarcity and its indicators at the zonal level. The graphs show the monthly variation of precipitation, green water flow, and green water storage on the primary axis and green water scarcity on the secondary axis.

On the other hand, the intra-annual variability in the green water scarcity showed further variability (Figure 13). Almost all the zones became an environmental hotspot during summer and beginning of fall (from June to August). This might be attributed to more crop growth and, hence, higher evapotranspiration in these months. In the worst case, the green water scarcity approached a value of 3 (300%). Such monthly and seasonal analyses of green water scarcity can be helpful for planning agriculture and for using available water in a more sustainable manner.

## 4. Conclusions

In this study, a hydrological modelling framework, using the soil and water assessment tool (SWAT), was applied to evaluate the spatial and temporal variability of blue and green water resources and water security in the Grand River watershed (GRW), Canada. The SWAT model of the GRW incorporated both natural and human factors influencing the hydrology of the watershed. Model calibration, validation, and uncertainty analyses showed 'good' to 'very good' quality of streamflow results. With the calibrated and validated model, we made further analyses on blue and green water resources and security. Overall, we arrived at the following conclusions:

- Spatial and temporal variability in the blue water resources can be explained by variability in climate factors (i.e., precipitation). Long-term analyses showed that the western sub-basins of the GRW had the highest blue water resources.
- The green water flow, on the other hand, was found to be associated with temperature, precipitation, and land use/land cover. The higher the temperature and more intense the agriculture was, the higher the green water flow was, as was observed for southern sub-basins of the GRW. Green water storage was found to be associated with various soil properties and was seen to be influenced by elevation or depth of soil. Thereby, a higher green water storage in the northern part of the watershed was observed.
- Blue water security analysis showed contrasting results for different EFR methods used. The presumptive method was found to be the most restrictive and conservative when compared to others. The blue water was scarce in some regions of the GRW and was found to be severe in

specific periods, especially for the Eramosa and Speed river sub-watershed and in summer months, on account of the higher percentage of urban area, more water use, and less blue water availability.

- Green water security analysis showed that the basin had no severe green water scarcity, and that it was adequate enough for practicing rain-fed agriculture in spring and fall seasons. It was found to be the highest in the southern Grand River sub-watershed because of the higher temperature, larger green water footprint, and lower green water availability.

We believe that such a modelling approach can be used in other Canadian watersheds as well; however, because of the various uncertainties and empirical relations involved, there is some room for improvement, such as: (a) groundwater needs to be accounted for more effectively in blue water security estimation studies; and (b) water use data needs to more refined and is needed over longer time-spans in order to do inter-annual analyses of blue water and green water security. Available soil moisture data with good resolutions can also be used for calibrating green water resources. In shaping future work, these limitations will be effectively acknowledged, and similar analysis would be extended to future time periods.

**Supplementary Materials:** The following are available online at http://www.mdpi.com/2071-1050/11/7/1883/s1, Supplementary Figure S1: (**a**) Monthly variability in different water resources components in Zone A–D of the GRW (Blue water scarcity calculated using Presumptive Standards method), (**b**) Monthly variability in different water resources components in Zone E–H of the GRW (Blue water scarcity calculated using Presumptive Standards method), Supplementary Figure S2: Flow duration curves showing the observed and simulated monthly flows for six gauging stations, Supplementary Table S1: Attribution of blue water for environmental flow requirement using different methods.

**Author Contributions:** Conceptualization, P.D., R.R.; methodology, P.D., N.K.S., and B.K.; formal analysis, P.D., N.K.S., and B.K.; investigation, P.D., N.K.S., and B.K.; resources, P.D., N.K.S., B.K., R.S., and N.A.; data, R.R., P.D., N.K.S., B.K., P.K.G., R.S., and N.A.; writing—original draft preparation, N.K.S., B.K.; writing—review and editing, R.R., P.D., and P.K.G.; supervision, R.R., P.D.; funding acquisition, P.D.

**Funding:** The funding for this study is provided by NSERC-Discovery Grant.

**Acknowledgments:** We would like to thank Jun Hou, Fahimeh Jafarianlari, and Harshpinder S. Brar for their insights and comments, which really helped to shape the manuscript.

**Conflicts of Interest:** The authors declare no conflict of interest.

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
