# Peer review of "Water Security Assessment of the Grand River Watershed in Southwestern Ontario, Canada"

_sustainability, doi:10.3390/su11071883_

Round 1

Reviewer 1 Report

The article titled “Water security assessment of Grand River Watershed in South-Western Ontario, Canada” discusses about water security of blue and green water based on SWAT model simulations of GRW. This paper offers ways in which water can be optimally allocated on a large scale using watershed modelling, which would be significant in the field of hydrology. Please find some suggestions in the attached file. 

Author Response

“ Water security assessment of Grand River Watershed in South-western Ontario, Canada ”, by Baljeet Kaur, Narayan Kumar Shrestha, Prasad Daggupati, Ramesh P. Rudra, Pradeep K. Goel, Rituraj Shukla, Nabil Allataifeh

General Comment from Authors and highlights of revision:

Following the suggestion of the Reviewers, we have done a few, minor revisions of the manuscript. We have diligently responded to all of the reviewer concerns in this version of the manuscript, whose responses are stated below. Doing so, we have restructured the few sentences and inserted new figures, which we believe, have improved the manuscript. We have:

Restructured the introduction and added few statistics in the abstract

Inserted a figure, in supplementary material for explaining the validity of calibration

Inserted another figure depicting the land-use and soil depth variability in the watershed. The map (Figure 2b) showing the soil depth was moved from supplementary material to the main text.

Reviewer 1

Water security assessment of Grand River Watershed in South-western Ontario, Canada

Dear authors,

I have reviewed this article and my recommendations for you are as follows:

Rev 1 Comment 1: For figures 1 and 2, I recommend that you put the cardinal orientation to the map (on the other maps you made this operation)

Reply: We have done the corrections. Thanks.

Rev 1 Comment 2: Line 252 it is not necessary to write Eqn. next to the equation itself, so only the number of the equation (1) will be written.

Reply: Thanks. We modified according to the suggestion.

Rev 1 Comment 3: The same will apply to all the equations presented.

Reply: Thanks.

Rev 1 Comment 4: Line 319 needs to be completed as: "... estimated as (Eqn. 7):"

Reply: Thanks. We added according to the reviewer’s suggestion.

Rev 1 Comment 5: Line 400 .... 425 mm / year (you should leave a space between the value and the unit of measure)

Rev 1 Comment 6: Line 401 .... 575 mm / year (you should leave a space between the value and the unit of measure)

Reply: We did the correction. Thanks.

Rev 1 Comment 7: Line 412 Put the correct number of the quoted figure (Figure 8 & Figure S2). What is the figure S2???

Rev 1 Comment 8: Line 437 Put the correct number of the quoted figure (Figure S1, a & b). What is figure S1???

Rev 1 Comment 9: Line 447 Put the correct table number (Table S1). What is Table S1???

Reply: Figures S1a, b and Figure S2 are in the supplementary material. Figure S1a and b explain monthly variability of different water resources components in all the zones of the GRW (Blue water scarcity calculated using Presumptive Standards method). Figure S2, is a new addition, shows the flow duration curves at all six calibration stations.

Rev 1 Comment 10: Line 503 Reformulate the title of Figure 11

Reply: We reformulated it. Thanks.

“Figure 12: Spatial distribution of Average Annual Green Water Scarcity” [Line 498]

General Recommendation: In my opinion, the introduction is a little too extensive and could be more restricted to easily browse the article.

Reply: Following the reviewer’s suggestion, we have deleted two general paragraphs from introduction.

Reviewer 2

The paper assesses blue and green water resources in the Grand River Watershed. The issue is properly developed, well written and adequately addressed. Notwithstanding, I think that some modifications should be made:

Rev 2 Comment 1:

Abstract: Authors should better introduce the subject to reader; the concept of water security is not previously described. Furthermore, specific values obtained should be also shown. Focus on main findings and the interest for the research community.

Reply: We have added a sentence introducing the ‘water security’ in the very beginning.

“Water security is the capability of a community to have adequate access to sufficient quantity and good quality water, and to safeguard resources for the future generations.”[Line 10-11]

We have also added some specific values at appropriate places.

“Model results showed that the SWAT model could simulate streamflow dynamics of the GRW with ‘good’ to ‘very good’ accuracy with an average Nash-Sutcliffe Efficiency, R2, and PBIAS being 0.75, 0.78 and 8.23%, respectively.” [Line 20-22]

“While both green (0.4-1.1) and blue water scarcity (0.25-2.0) showed marked temporal as well as spatial variability, the blue water scarcity was found to be the maximum in urban areas on account of higher water use and less blue water availability.” [Line 25-28]

Rev 2 Comment 2:

Line 20: List all methods.

Reply: Thanks. We listed all the methods used for estimating EFR in the abstract.

“In particular, while calculating blue water scarcity, three different methods were used in determining the environmental flow requirement, namely, the presumptive standards method, the modified low stream-flow method and the variable monthly flow method.” [Line 17-20]

Rev 2 Comment 3:

Rev 2 Comment 4:

Lines 25-27: Specify how could be helpful.

Reply: We modified the very last sentence of the abstract according to reviewer’s suggestion.

“We believe that know-how of green and blue water security situation would be helpful in sustainable water resources management of the GRW and identifying hotspots which need immediate attention.” [Line 29-31]

Rev 2 Comment 5:

Introduction:

-Lines 70-71: Use abbreviation BW and GW (blue water and green water)

Reply: We updated it according to the reviewer’s suggestion. Thanks.

Rev 2 Comment 6:

- Line 101: what does "7.6% of its area under water resources" mean?

Reply: We modified the sentence. In place of “its area” we added “total area”. Thanks. [Line 88-89]

Rev 2 Comment 7:

- Lines 120-131. Define the application of the study in other areas and the…

Reply: We added a sentence in the last paragraph of the introduction.

“Such modelling approach can also be used in other cold-climate, snow-dominated regions, which are assumed to be water-abundant but are actually water-scarce.” [Line 116-118]

Rev 2 Comment 8:

- References do not follow journal rules. Please revise.

Reply: We revised the references according to journal’s rules. Thanks.

Rev 2 Comment 9:

Materials and methods:

- Lines 139-141: I miss data and/or statistics.

- Line 149: same as above, same as above

Reply: We added few statistics representing the average annual precipitation and temperature.

“The weather in the GRW is moderate to cool temperate (average annual precipitation ranging between 800-900 mm and temperature between 8-10°C [33]) and the watershed experiences four main seasons including winter, which is cool and dry, and summer, which is hot and humid.” [Line 129-132]

Rev 2 Comment 10:

- Figure 2: reformulate caption, confusing.

Reply: Thanks. We reformulated the caption of Figure 3 (in the revised version). Figure 3 was Figure 2 in the previous version.

Rev 2 Comment 11:

- Lines 161-162: add references.

Reply: We added the reference. Thanks.

Rev 2 Comment 12:

- Lines 228, 230: follow journal´s rules for references citing.

Reply: We modified the references according to the journal’s style.

Rev 2 Comment 13:

- Lines 267: after "...security studies" write only the number of references, not the authors.

Reply: We changed the referencing style according to the reviewer’s suggestion.

Rev 2 Comment 14:

- Lines 285-287: consider reformulate. Note clear.

Reply: Thanks. We reformulated it and enlisted the three classes of flow.

“It classifies the flow into three classes:

·         low flow (MMF ≤ 40% of the mean annual flow (MAF))

·         intermediate flow (40% of MAF < MMF < 80% of MAF)

·         high flow (MMF> 80% MAF)

where, MMF stands for Mean Monthly Flow for a particular month.” [Line 278-283]

Rev 2 Comment 15:

Results and discussion: 

- Line 327: Replace "has been" by "is"

Reply: We replaced it. Thanks.

Rev 2 Comment 16:

- Lines 328-331: Discuss values obtained.

Reply: We don’t understand the comment. These are not the ‘values obtained’, rather the ‘values used’ as upper and lower range for sensitivity analysis. We don’t think there is a need to elaborate the range of values used.

Rev 2 Comment 17:

- Line 333: Add references.

Reply: Thanks. We added the references. Line 329-330 in the revised version.

Rev 2 Comment 18:

- Figure 4: Consider replace by flow-duration-curves. I think current graphics are difficult to interpret and to discuss. ??

Reply: While we agree with the comments, we believe that time series plots with 95% PPU bands on simulated results also convey meaningful message. Hence, we added the flow-duration-curve figure as supplementary material. Please refer S2.

Rev 2 Comment 19:

- Lines 341-342: Reformulate. Confusing.

Reply: We changed few words in the sentence. Thanks. Line 337-339 in the revised version.

Rev 2 Comment 20:

- Line 343: "...value of 1.5", add references.

Reply: We added a reference. Thanks.

Rev 2 Comment 21:

- Table 3: I miss a deeper analysis of Unsatisfactory (U) results and a correlation with other conclusions.

Reply: As depicted, we performed multi-site calibration and validation using NSE as the objective function. In such a process, it is common that goodness of fit statistics are better at one station than others. We consider Grand at Brantford as main station, as it the located at the most downstream reach. Furthermore, most of the ‘unsatisfactory’ results are obtained during validation period, and this is a very common issue too. A model may not perform as well as in calibration period as climatic forcing will always be slightly different in validation period (Lines: 357-360). Furthermore, ‘unsatisfactory’ rating against all three goodness of fit statistics is obtained only for a station and during validation period (Table 3). Hence, we don’t see a consistent pattern on it.

Rev 2 Comment 22:

- Line 176: Figures 6 and 7

Reply: We corrected it. Thanks.

Rev 2 Comment 23:

- Lines 376-379: Consider reformulate.

Reply: Thanks. We reformulated as follows:

“While blue water yield showed an increasing trend using the Sen’s slope analysis, a kind of cyclic pattern with increasing values in some years (positive Z-value, Figure 8 in years 1975-1980) and decreasing (negative Z-values, Figure 8 in years 1960-65) in others is evident.”[Line 373-376]

Rev 2 Comment 24:

- Line 386: "...thicker band", how much?

Reply: We added a sentence to the parenthesis to highlight the band thickness.

“It should be noted that a thicker band (with band thickness varying from 200 mm to over 400 mm) was obtained for blue water resources when compared to the same for green water resources, which implied higher variation in the blue water resources and hence higher uncertainty.” [Line 383-386]

Rev 2 Comment 25:

- Line 448: Replace "was" by "were"

Reply: We believe that the sentence is correct.

Rev 2 Comment 26:

References: follow journal´s rules.

Reply: Thanks. We modified the references according to the journal rules.

Reviewer 3:

The article titled “Water security assessment of Grand River Watershed in South-Western Ontario, Canada” discusses about water security of blue and green water based on SWAT model simulations of GRW. This paper offers ways in which water can be optimally allocated on a large scale using watershed modelling, which would be significant in the field of hydrology. Therefore, I propose to ACCEPT the manuscript for publication in Sustainability journal after addressing some minor issues as suggested below:

Overall, the manuscript reads well; however formatting and language corrections is needed throughout the manuscript. ??

Reply: Thanks. We have now undergone a thorough proof-reading of the manuscript.

Other minor corrections:

Rev 3 Comment 1:

Line 121: Change “built-up” to build model or rewrite the sentence

Reply: Thanks. We changed “built-up” to “build”.

Rev 3 Comment 2:

Line 122: There is no mention of SWAT before this line. Please describe few sentences about different models used for this purpose in Introduction

Reply: We added a sentences to highlight the approached used to estimate water scarcity.

“Also, various modelling pathways have been used by the researchers to assess water resources, ranging from Global Hydrological Models (GHMs) like, WaterGap 3.0 [15] to Catchment-scale Hydrological Models (CHMs), like Soil and Water Assessment Tool (SWAT) [16].” [Line 73-76]

Rev 3 Comment 3:

Figure 1: What area the red lines in the Inset figure? Legend for all items shown in the figure? Brown in inset is GRB? Then the boundary of the main GRB figure should be brown.

Reply: We modified Figure 1 according to reviewer’s suggestion.

Rev 3 Comment 4:

Table 1: What is the sub-basin-wide land-use distribution in the study area?

Reply: We added/modified a sentence explaining land-use distribution in the study area. Also, we added the land-use map (Figure 2).

“Maximum area in the GRW is under agriculture (43%), followed by pastures and range-grasses (26.92), forests (12%) and small fragments of urban areas (9.29%) and wetlands (1.8%) (Figure 2).” [Line 127-129]

Rev 3 Comment 5:

Line 208: Precipitation was from 1950; but total time period selected was from 1967. Why?

Reply: We corrected the year to 1950. Also, we added a sentence to explain why we did not start calibration from 1950. Thanks.

“As such, the total time period (1950-2015) was divided into warm-up period (1950-1952), validation periods (1970-1981 and 2008-2015), and calibration (1982-2007) period. As, streamflow data for the period 1953-1969 was not available, so calibration and validation was done in the period 1970-2015.” [Line 200-203]

Rev 3 Comment 6:

Line 234: Observed or SWAT output? IF so, which one?

Reply: We used output obtained from SWAT to estimate the trend of various components using Sen’s slope.

Rev 3 Comment 7:

Lines 314 – 320: The available water is taken as the plant available water, which is water content at FC - water content at WP. So, GWavailable is taken as this FC-WP at end of each time step?

Reply: Yes, GWavailable is the amount of available soil water at the beginning of the time step, which is equal to moisture content at FC-WP. In SWAT output files it can be found under the column SW_INIT.

Rev 3 Comment 8:

Lines 349-351: Please explain as pertaining to the study area and its model set-up. Moreover, only two sub-basins show less accuracy in validation, which may be due to its specific hydrologic characteristics. Please evaluate.

Reply: We totally agree. Site-specific validation is indeed preferred. It requires investigation right from the observed data, e.g. adequacy of rating curve used to derive streamflow. The message that we wanted to convey in these lines, however, was that “a lower accuracy was indeed expected in validation period”. The fact is well established in hydrological community, as such we endeavor to include the phrase “due to obvious reasons”.

Rev 3 Comment 9:

Line 375: I thought the analysis period started at 1965 (Refer: Line 208) Please check.

Reply: Thanks. We corrected the year to 1950 in Line 199-202.

Rev 3 Comment 10:

Lines 415-417: This sentence is not needed. Please remove.

Reply: We removed it. Thanks.

Rev 3 Comment 11:

Line 426: Not always! It is dependent on soil hydraulic characteristics.

Reply: We agree. We added a phrase “in general” to further generalize the sentence.

Rev 3 Comment 12:

Figure 11: Specify annual water scarcity in figure caption.

Reply: Thanks. We modified it according to the reviewer’s suggestion.

Rev 3 Comment 13:

Line 533: Change “The blue was scarcity” to “The blue water was scarce”

Reply: Thanks. We modified the sentence according to reviewer’s recommendation.

Rev 3 Comment 14:

Figure S2: This figure could be at the main text along with land-use map.

Reply: Thanks. We moved the figure into the main text and also added a land-use map along with it (Figure 2).

We would like to thank the reviewers for their careful consideration of the manuscript. We have addressed all of the reviewer’s concern in the revised manuscript.

Reviewer 2 Report

Dear Authors,

I have attached the review. I have a few suggestions to improve the article.

Author Response

(The authors gave the same response as above.)

Reviewer 3 Report

The paper assesses blue and green water resources in the Grand River Watershed. The issue is properly developed, well written and adequately addressed. Notwithstanding, I think that some modifications should be made: 

Abstract: Authors should better introduce the subject to reader; the concept of water security is not previously described.Furthermore, specific values obtained should be also shown. Focus on main findings and the interest for the research community. 

-Line 20: List all methods.

-Lines 25-27: Specify how could be helpful.

Introduction:

- Lines 70-71: Use abbreviation BW and GW (blue water and green water)

- Line 101: what does "7.6% of its area under water resources" mean?

- Lines 120-131. Define the application of the study in other areas and the

- References do not follow journal rules. Please revise.

Materials and methods:

. Lines 139-141: I miss data and/or statistics.

- Line 149: same as above,

- Figure 2: reformulate caption, confusing.

- Lines 161-162: add references.

- Lines 228, 230: follow journal´s rules for references citing.

- Lines 267: after "...security studies" write only the number of references, not the authors.

- Lines 285-287: consider reformulate. Note clear.

Results and discussion: 

- Line 327: Replace "has been" by "is"

- Lines 328-331: Discuss values obtained.

- Line 333: Add references.

- Figure 4: Consider replace by flow-duration-curves. I think current graphics are difficult to interpret and to discuss. 

- Lines 341-342: Reformulate. Confusing.

- Line 343: "...value of 1.5", add references.

- Table 3: I miss a deeper analysis of Unsatisfactory (U) results and a correlation with other conclusions.

- Line 176: Figures 6 and 7

- Lines 376-379: Consider reformulate.

- Line 386: "...thicker band", how much?

- Line 448: Replace "was" by "were"

References: follow journal´s rules.

Author Response

(The authors gave the same response as above.)
